# Recent Advances in Surface Plasmon Resonance (SPR) Technology for Detecting Ovarian Cancer Biomarkers

**DOI:** 10.3390/cancers15235607

**Published:** 2023-11-27

**Authors:** Vikneswary Ravi Kumar, Nirmala Chandralega Kampan, Nor Haslinda Abd Aziz, Chew Kah Teik, Mohamad Nasir Shafiee, P. Susthitha Menon

**Affiliations:** 1Department of Obstetrics and Gynaecology, Faculty of Medicine, Universiti Kebangsaan Malaysia, Kuala Lumpur 56000, Malaysiadrchewkt@gmail.com (C.K.T.);; 2Institute of Microengineering and Nanoelectronics (IMEN), Universiti Kebangsaan Malaysia (UKM), Bangi 43600, Malaysia

**Keywords:** epithelial ovarian cancer (EOC), biomarkers, cancer antigen 125 (CA125), human epididymis protein 4 (HE4), immunoassay, multiplex, surface plasmon resonance (SPR)

## Abstract

**Simple Summary:**

This review significantly contributes to the research community by providing insights into the role of serum-based biomarkers for the diagnosis of ovarian cancer. It also provides a comprehensive overview of recent advancements in immunoassay detection and employing multiplex technology and SPR biosensors to identify CA125 and HE4 biomarkers. Furthermore, it addresses the challenges associated with these diagnostic methods. This valuable information not only enhances our understanding of ovarian cancer diagnosis but also serves as a reference point for researchers and clinicians working in the field, facilitating further advancements and improvements in early detection methods.

**Abstract:**

Epithelial Ovarian Cancer (EOC) is a leading cause of cancer-related deaths among women, mainly due to a lack of early detection and screening methods. Advanced immunoassay techniques, such as Luminex and proximity extension assay (PEA) technology, show promise in improving EOC detection by utilizing highly sensitive and specific multiplex panels to detect multiple combinations of biomarkers. However, these advanced immunoassay techniques have certain limitations, especially in validating the performance characteristics such as specificity, sensitivity, limit of detection (LOD), and dynamic range for each EOC biomarker within the panel. Implementing multiplexing in point-of-care (POC) biosensors can enhance EOC biomarker detection, with Surface Plasmon Resonance (SPR) being a versatile option among optical biosensors. There is no study on multiplex SPR biosensors specifically tailored for diagnosing EOC. Recent studies have shown promising results in the single detection of EOC biomarkers using SPR, with LOD for cancer antigen 125 (CA125) at 0.01 U/mL^−1^ and human epididymis protein 4 (HE4) at 1pM. This study proposes a potential roadmap for scientists and engineers in academia and industry to develop a cost effective yet highly efficient SPR biosensor platform for detecting EOC.

## 1. Introduction

Epithelial Ovarian Cancer (EOC) is a form of cancer that originates in the outer layer of cells on the ovary’s surface. It ranks as the fifth most prevalent cancer and is a significant cause of death [1,2]. EOC accounts for 90% of all ovarian cancer and displays significant variation in terms of its appearance, clinical characteristics, molecular makeup, histologic subtype, and chemotherapy sensitivity; therefore, it has an impact on the prognosis of ovarian cancer [3,4,5,6]. EOC primarily manifests in older women, with the average age of diagnosis being 63 years, compared to younger women [7]. Despite a lack of screening methods, the rise in survival rates for ovarian cancer is largely credited to aggressive surgical debulking that includes hysterectomy and bilateral salpingo-oophorectomy, a standard surgical treatment for EOC and often combined with chemotherapy, specifically the conventional carboplatin-paclitaxel regimen [6,8]. Recently embraced treatments, specifically Poly ADP Ribose Polymerase (PARP) inhibitors, have proven to be effective for women with EOC, especially those with BRCA mutations [5,9]. Despite the initial effectiveness of these inhibitors, many patients eventually undergo an elevation in deoxyribonucleic acid (DNA) damage, resulting in resistance to the PARP inhibitors and, consequently, an elevated risk of mortality [9].

American Cancer Society estimates for ovarian cancer in the United States for 2023 approximately 19,710 new cases in 2023 (1% of all new cancer cases) and the estimated number of deaths in 2023 is 13,270 (2.2% of all cancer-related deaths) [10,11]. Despite these advances, the Surveillance, Epidemiology, and End Results (SEER) data indicate that the 5-year relative survival rate for ovarian cancer from 2013 to 2019 is at 50.8% [10]. In contrast, a small percentage of patients are diagnosed with stage I EOC and 78% of women diagnosed at this stage survive for at least one year [8,12]. Hence, timely identification of early-stage epithelial ovarian cancer (EOC) is pivotal for enhancing prognosis. It is essential to address the non-specific clinical signs of EOC and discover biomarkers capable of signaling the existence of preclinical or early-stage tumors, while also offering prognostic insights [4,13].

In the last twenty years, substantial endeavors have been devoted to evaluating the effectiveness of cancer antigen 125 (CA125), a tumor marker, and ultrasound scanning as a means of screening for ovarian cancer [14,15,16]. Noteworthy advancements have been achieved in enhancing the performance of these tests. Presently, there are ongoing extensive trials aiming to evaluate the effects of screening the general population. The development of serum proteomic analysis has generated significant interest in investigating novel combinations of serum protein markers that offer high sensitivity [17].

Immunoassays play a crucial role in the investigation of antigen–antibody interactions, enabling the sensitive detection of EOC biomarkers [18]. These assays are valued for their high specificity and sensitivity, which contribute to the identification of reliable biomarkers for diagnosing and prognosing EOC [19,20]. In clinical diagnostics, the most commonly used conventional immunoassays, such as Enzyme-Linked Immunosorbent Assay (ELISA), Chemiluminescence Immunoassay (CLIA), and Electrochemiluminescence Immunoassay (ECLIA), are widely employed [13]. However, label-free tests like lateral flow immunoassay (LFIA) are less sensitive than ELISA and are unable to provide accurate results [20,21]. It is important to note that these diagnostic tools, utilized for measuring the serum concentration of potential biomarkers, require expensive equipment, considerable time, plenty of consumables, and a large volume of samples [21,22,23].

Conventional immunoassay tests face a challenge in accurately identifying a particular analyte while minimizing background interference and achieving sensitive detection. To address this limitation, various immunoassay methods have been employed to detect the analyte. However, these methods often lack the necessary sensitivity to detect proteins found in low concentrations, including CA125 and human epididymis protein 4 (HE4) [19,20,22]. The current approaches for investigating EOC biomarkers linked to exosomes in human serum include flow cytometry, protein microarray, diagnostic magnetic resonance, nanoplasmonic sensing technology, and microfluidics [24]. In recent years, multiplex immunoassay techniques have gained popularity for simultaneously detecting multiple combinations of EOC biomarkers. Nonetheless, these techniques present specific challenges due to the simultaneous analysis of multiple markers.

On the other hand, optical biosensors have gained popularity due to their ability to offer multiplexed detection and specifically recognize and bind target analytes. Especially, Surface Plasmon Resonance (SPR) is a widely used optical biosensor platform that enables label-free measurement of biomolecular interactions [25,26,27]. It allows for the real-time detection of biomolecules and offers greater versatility compared to other optical biosensors [25]. Despite these advancements, there is currently no existing study that has developed a multiplex SPR biosensor specifically for diagnosing EOC using human serum. Nonetheless, this study highlights the potential of utilizing SPR biosensors to simultaneously detect CA125 and HE4 biomarkers in EOC.

The objective is to discover a SPR (Surface Plasmon Resonance) method that can effectively detect low concentrations of CA125 and HE4 while offering a wider dynamic range for these biomarkers. Currently, there is a lack of research that specifically focuses on developing a multiplex SPR biosensor for diagnosing EOC using human serum. Hence, this review carries the potential to advance the development of a multiplex SPR biosensor, which would hold significant value in healthcare, particularly for monitoring EOC biomarkers. Additionally, this review’s findings are important in guiding the selection of appropriate SPR instruments and immobilization techniques to construct a tailored multiplex platform suitable for specific clinical applications.

In summary, this review article provides a comprehensive overview of current advancements in multiplexed detection of EOC biomarker panels, focusing on the challenges and performance of SPR in detecting CA125 and HE4 biomarkers. The study emphasizes the need to determine low limits of detection (LOD) and wider dynamic ranges for CA125 and HE4 in developing multiplex SPR biosensors and challenges as well.

## 2. Serum-Based EOC Biomarkers

In human blood circulation, commonly present are proteins and genes (cell-free DNA) which form biomarkers that comprise a spectrum of bioactive compounds such as Growth Factors, Glycoproteins, Inflammatory Cytokines, Enzymes, and Exosomes [25,28]. These are produced by tumor cells and play a significant role in the tumor microenvironment (TME) to promote cancer [6]. Figure 1 shows the interactions between tumor cells and immune cells in TME components and unidirectional communication between circulating proteins and exosomes. Presently, existing evidence indicates that the measurement of cell-free DNA in epithelial ovarian cancer (EOC) yields superior results and demonstrates diagnostic capabilities similar to protein biomarkers. However, the diagnostic value of cell-free DNA for EOC patients remains uncertain, necessitating comprehensive large-scale prospective studies for robust validation [4,29].

Recent studies underscore the importance of placing greater emphasis on exosomal MicroRNA (miRNA), which is extensively implicated in the development of the epithelial ovarian cancer tumor microenvironment (EOC-TME). This emphasizes its crucial role in both the initiation and progression of tumors [30,31]. Several researchers have examined the expression profiles of miRNAs in serum samples from individuals with EOC, with the goal of identifying biomarkers for the condition [30,32]. Although a specific study effectively identified certain circulating miRNAs consistently linked to epithelial ovarian cancer (EOC), there is a lack of consensus regarding the methodologies employed [32], and the costs associated with miRNA profiling are substantial [24]. Currently, there is no definitive biomarker available to predict the response to therapy, whether based on mRNA, protein, or DNA. None of these markers have received approval for clinical use [5,33]. Therefore, it is recommended to lean towards circulating serum-based biomarkers for diagnosing EOC, while also considering potential factors that could impact the accuracy of an EOC diagnosis.

Serum-based biomarkers are commonly the most effective for detecting circulating proteins when using the immunoassay technique. They are often the preferred choice for identifying biomarkers due to their ability to provide specific information with minimal interference from background [15]. Serum-based biomarkers are considered the optimal choice for predicting EOC diagnosis, prognosis, and treatment effectiveness. They enable the detection of tumor-related biomarkers that display quantitative kinetic and affinity-binding properties [34,35]. However, the strategy for biomarker detection depends on factors such as the characteristics of the protein(s) of interest, the specific immunoassay technique employed, and the nature of the research.

Although numerous serum-based biomarkers have been discovered and evaluated in the past decade, only the CA125 and HE4 biomarkers have been proven to be effective in predicting different types of epithelial ovarian cancer [36,37]. Currently, the CA125 and HE4 biomarkers are utilized independently for the diagnosis of EOC, as both biomarkers heavily rely on the stage of the disease and can vary in their ability to classify patients into low or high-risk groups [4,38,39]. Consequently, relying solely on the detection of a single biomarker in serum for the diagnosis of EOC is insufficient to make accurate predictions in clinical settings.

## 3. Conventional Diagnostic EOC Biomarker in Immunoassay Technique

Several serum-based biomarkers have been identified as crucial for the development of effective approaches to early detection of EOC. These approaches are essential for broad clinical applications in diagnosis, prognosis, and observation [6,40]. When it comes to immunoassay methods for EOC diagnosis, it is expected that they achieve a minimum specificity (SP) of 99.6% and a sensitivity (SN) of over 75% for early-stage EOC [16]. This is necessary to overcome unacceptable false-positive results and achieve a positive predictive value of 10% [36]. Validation studies and performance evaluations of biomarkers and immunoassay methods play a critical role in determining their accuracy and reliability for the diagnosis and screening of EOC.

In the detection of EOC, FDA (Food and Drug Administration)-approved single biomarkers such as CA125, HE4, Carbohydrate antigen 19-9 (CA19-9), and Carcinoembryonic antigen (CEA) are commonly measured in serum using cutoff values derived from the biomarker’s range or predefined ranges [41,42]. In Table 1, we provide a summary of FDA-approved biomarkers and select serum samples based on their SP% and SN% to predict the molecular weight (MW) of EOC. The concentration of individual biomarkers is typically expressed in units such as U/mL, U mL^−1^, or µg mL^−1^ for CA125, CA19-9, and CEA, and pM (picomoles/L) for HE4 [43]. However, among these biomarkers, CA125 and HE4 are the most widely used and well-known, playing a crucial role in diagnostic tests. It is important to note that the CA125 cutoff value has low SP% due to elevated levels in various physiological circumstances such as menstruation, pregnancy, and pre- and post-menopause [38]. On the other hand, the HE4 biomarker concentration exhibits high specificity in the serous EOC subtype [44]. Nevertheless, relying solely on the specificity of a single detection biomarker may not provide reliable results [45].

The diagnostic effectiveness of CA19-9, CA125, and CEA in preoperative serum was assessed through ELISA, ECLIA, and Magnetic Bead assay methodologies, highlighting their utility in predicting and distinguishing various types of tumors [46]. Moreover, CA125, HE4, and CA19-9 play a role in prognosticating patients with epithelial ovarian cancer (EOC). A meta-analysis, encompassing 23 studies with a total of 10,594 women diagnosed with EOC, revealed that a higher pre-treatment serum CA125 level, independent of FIGO stages and treatments, was correlated with poorer overall survival (OS: HR = 1.62, 95% CI = 1.270–2.060, *p* < 0.001) and progression-free survival (PFS: HR = 1.59, 95% CI = 1.44~1.76, *p* < 0.001). Consequently, serum CA125 stands out as a reliable indicator for predicting the risk of EOC disease progression [47]. Additionally, preoperative HE4 emerges as a promising prognostic biomarker in EOC, especially in serous tumors, with elevated preoperative plasma HE4 levels (≥277 pmol/L) showing a significant association with increased EOC mortality (adjusted hazard ratio (aHR): 1.90; 95% CI: 1.09–3.29) [48].

Furthermore, Rong and Li discovered that the timely normalization of HE4 and CA125 levels in the early stages could serve as an indicator for platinum response and prognosis in ovarian cancer patients. Thus, monitoring these biomarkers in combination throughout the initial chemotherapy phase could provide valuable insights into predicting platinum sensitivity and the likelihood of progression and relapse [49] Another study uncovered the role of CA19-9 as a prognostic biomarker. Zhu et al. found that combining postoperative CA19-9 and CA-125 appeared to have significant clinical value for prognosis in patients with ovarian clear cell carcinoma (OCCC) after initial debulking. Elevated postoperative CA19-9 was identified as an independent risk factor for both recurrence-free survival (RFS: HR, 5.0; *p* = 0.005) and overall survival (OS: HR, 1.1; *p* = 0.035) in patients with normal postoperative CA-125 levels [50].

Although CA125, HE4, and CA19-9 are standard detection markers for prognosis, there are a few new biomarkers used to predict the progression of disease in advanced EOC including Bikunin, Osteopontin (OPN), and Creatine Kinase B (CKB) [5]. These biomarkers are predominantly utilized in clinical diagnosis and prognosis, aiding in predicting the disease stage and monitoring chemotherapy response through ELISA. A recent review has outlined nearly two hundred prognostic biomarkers linked to EOC, establishing a robust groundwork for the exploration of innovative treatment approaches [33,51]. The identification of prognostic factors in EOC patients is vital for devising optimal treatment strategies [33]. This involves establishing an algorithm to monitor various target cohorts, necessitating extensive databases.

To enhance the inherent characteristics of biomarkers, Multivariate Index Assays have been developed [36]. These tests integrate results from multiple analytes to generate a risk score for predicting the presence of disease. In Table 2, we provide an overview of a few screening strategies, including the Risk of Ovarian Malignancy Algorithm (ROMA), which combines CA125 and HE4; the Risk of Malignancy Index (RMI), calculated by multiplying the risk obtained from transvaginal ultrasound with the CA125 value (U/mL); OVA1; and Overa (the second generation of OVA1 test). Three tests based on this principle, namely OVA1, Overa, and ROMA, have received approval from the U.S. Food and Drug Administration [52].

Pooled estimates for the ROMA index demonstrate a SN of >85 to 97% and a specificity of >80 to 91% [15,40,53,54,55]. However, another study reported that ROMA did not provide additional clinical benefits compared to the detection of CA125 or HE4 [42]. In 1990, Jacobs et al. established the RMI using a cutoff value of 200, which exhibited increased SN (>87 to 97%) and specificity (>80 to 90%) compared to assessing CA125 levels alone [38,54,56]. In a prospective clinical study, OVA1 and Overa proved to be useful when combined with pre-surgical clinical and radiological assessments or physician evaluations, aiding in the triage of women at high risk of EOC for referral to a gynecological oncologist before undergoing surgery [36,45].

Multivariate Index assays are not designed to be definitive diagnostic tests but rather serve as triage or referral tests. Some biomarkers incorporated in multivariate algorithms for EOC may lack sufficient specificity to reliably detect the disease. Moreover, the lack of S_P_ these biomarkers to a particular tumor or tissue type, along with their presence in non-cancerous conditions, limits their suitability for screening and early-stage diagnosis [43,45,56,57,58]. Additionally, the widely used single detection biomarkers, CA125 and HE4, employ ECLIA and magnetic-bead assay detection methods. The heterogeneity of EOC, including its various subtypes and molecular profiles, further complicates the accurate detection of specific subtypes using these methods [44].

As a result, the quest for biomarkers with increased S_P_ and accuracy remains an ongoing challenge. The accuracy and reliability of EOC detection techniques can be enhanced through the development of more precise and sensitive biomarkers, improved multivariate analytical algorithms, and standardized procedures. Combining multiple FDA-approved serum-based EOC biomarkers with emerging multiplex technologies may present new opportunities for improving EOC diagnosis [39]. A challenge in biomarker verification/validation lies in target cohort selection and the development of suitable platforms that are both precise and have high throughput as cohort sizes increase. Antibody-based immunoassays have traditionally been employed for biomarker verification, but the singleplex nature of these assays necessitates individual testing for each analyte [59]. In contrast, multiplex assays offer the potential for obtaining more reliable quantitative information through highly parallel analysis, compared to single analyte ELISA methods.

**Table 1 cancers-15-05607-t001:** List of FDA-approved single biomarkers for detection of EOC.

Criteria	Value/ Target Cohort	Biomarkers			
		CA125 U/mL	HE4 pM/L	CA19-9 U/mL	CEA ng/mL
Type		Mucinous Glycoprotein	Glycoprotein	Mucinous Glycoprotein	Glycoprotein
Diagnostic		✓	✓	✓	✓
Prognosis		✓	✓	✓	
Monitoring				✓	✓
FDA Approval (Year)		1981	2008	2002	1985
Cut-off		35	>70 or 140	<37 or 27	2.5–5
M_W_ (kDa)		>200	25	1000	180
Immunoassay Technique: ECLIA/Magnetic bead assay	S_P_%	>60–80	>96–100	79.01	>88–100
	S_N_%	>60–96	>63–83	35.71	>38–66.3
	Benign	9.02–54.92	49.01–54.49	19.15	2.71
	Malignant	368	245.9		
	Late-stage	184.62	234.77	45.61	9.27
Ref.		[60,61]	[60,61,62,63]	[36,54,64]	[52,64,65,66]

Abbreviations: CA 125: cancer antigen 125; HE4: human epididymis secretory protein 4; CA19-9: carbohydrate antigen 19-9; CEA: carcinoembryonic antigen; FDA: Food and Drug Administration; *M_W_*: Molecular weight; ECLIA: Electrochemiluminescence Immunoassay; S_P_%: Specificity; S_N_%: Sensitivity. ✓-Used as biomarker.

**Table 2 cancers-15-05607-t002:** List of FDA-approved biomarker-based multivariate Index assays for detection of EOC.

	ROMA Test	RMI	OVA1 Test	OVA2 (Overa)
FDA Approval (Year)	2011	N/A	2009	2016
Biomarker				
• CA 125	x	x	x	x
• HE4	x			x
• B2M			x	
• ApoA-I			x	x
• FSH				x
• TRF			x	
• Transferrin			x	x
Immunoassay technique	ECLIA [Cobas 8000, Roche Diagnostics Scandinavia AB, Sweden]// Magnetic bead assay [xMAP bead-based technology (Luminex, Austin, Texas)]		ECLIA Cobas 6000 (Roche, Germany) /IVDMIA [Quest Diagnostics Incorporated; Vermillion, (Austin, Texas)]	IVDMIA [Quest Diagnostics Incorporated; Vermillion, (Austin, Texas)]
S_P_ (%)	>80 to 91	>80 to 90	<75 to 26	28–35
S_N_ (%)	>85 to 97	>87 to 97	>78 to 98	77–96
Ref.	[40,52,60,65,67]	[40,52,60,65,67]	[36,44,52,54,60]	[52,60,66]

Abbreviations: ROMA: Risk of Ovarian Malignancy Algorithm; Risk of Malignancy Index (RMI); FDA: Food and Drug Administration; CA 125: cancer antigen 125; HE4: human epididymis secretory protein 4; B2M: β2-microglobulin; ApoA-I: Apolipoprotein AI; FSH: follicle-stimulating hormone; Transthyretin (TRF); ECLIA: Electrochemiluminescence Immunoassay IVDMIA: In Vitro Diagnostic Multivariate Index Assay S_P_: Specificity; S_N_: Sensitivity. N/A: Not applicable; no data show the FDA approval year for RMI. The ‘x’ symbol indicates the inclusion of a biomarker in the ROMA test, RMI, OVA1 test, and OVA3 (Overa).

## 4. Advanced Diagnostic Multiplex EOC Biomarker in Immunoassay Technique

### 4.1. Significance of Multiplexed EOC Biomarker

The importance of multiplexed EOC biomarker detection lies in its capacity to provide a more comprehensive understanding of the disease [68]. EOC and its responses to therapy involve intricate biological processes and proteins. Therefore, relying on the detection of a single biomarker may not offer a complete assessment of the disease’s status or treatment response [59]. Multiplexing allows for the simultaneous detection of multiple analytes, enabling researchers and clinicians to assess the interaction between different biomarkers and biological processes associated with EOC [16]. Multiplex immunoassays offer a more precise and comprehensive evaluation of the disease by measuring the levels of multiple biomarkers in serum samples and emphasizing the ability to predict the progression of EOC [53].

Traditionally, methods such as ELISA, ECLIA, magnetic bead assays, and polymerase chain reaction (PCR) have facilitated the detection of single analytes [68]. However, the development of multiplex immunoassays has opened new avenues for EOC detection and biomarker analysis. By simultaneously screening multiple biomarkers, multiplex tests have the potential to enhance predictive discrimination between groups, such as distinguishing between malignant and benign conditions. Both S_N_ and S_P_ have commonly employed measures to evaluate the accuracy of multiple biomarker panels, often summarized by the area under the receiver operating characteristic (ROC) curve, also known as the area under the curve (AUC) [69]. The ROC curves aid in categorizing patients as positive or negative based on test results and determining the appropriate cutoff value for diagnosis. An AUC value exceeding 0.7 indicates strong predictive discrimination between groups [70]. By simultaneously screening numerous biomarkers in a multiplex test, the AUC value can potentially be improved, thereby enhancing the diagnostic accuracy and predictive capability of the test.

### 4.2. Multiplex Panel Analysis via Current Labelled Immunoassay

In recent years, several studies have focused on exploring customized multiplex panels for the detection of EOC biomarkers. Muinao et al. conducted a study summarizing diagnostic approaches between 2003 and 2018, emphasizing the use of CA125 in combination with other biomarkers. Lokshin et al. developed a panel that included CA-125 along with CA 19-9, EGFR, G-CSF, Eotaxin, IL-2R, cVCAM, and MIF, which exhibited high SN (98.2%) and SP at 98.7% using the laboratory multianalyte profiling (LabMAP™) system [36,55]. The LabMAP™ system (Luminex Corporation, Austin, Texas) utilized microspheres with patented dyeing methods, allowing for covalent ligand attachment. Yurkovetsky et al. found that a combination of CA-125, HE4, CEA, and VCAM-1 achieved a specificity of 98% and a SN of 86% using Luminex Multi-Analyte Profiling (xMAP) technology [65]. However, the specificity level was still insufficient for screening the general population, as noted by Lokshin and colleagues [42,65].

Recent studies from 2018 to 2022 have focused on the utilization of multiplex panels with various combinations of EOC biomarkers using current labeled immunoassay methods (Table 3). Chen et al. employed an ECLIA panel, including CA125, to differentiate between benign and malignant cases, achieving a higher SN of 88.52% compared to CA125 and HE4 alone [65,66]. However, the combination of biomarkers did not significantly enhance diagnostic efficacy compared to the combination of HE4 and CA125 alone, as determined by the AUC analysis.

An emerging trend in immunoassays is the development of solid support versions utilizing microbeads, particularly magnetic beads with paramagnetic properties. These micrometer-sized magnetic beads allow for the removal of non-specifically bound serum-based biomarkers and are embedded in a non-magnetic/polymer matrix [57,58,71]. Examples of magnetic bead-based assays include the Luminex multiplex assay (Microplex, Microspheres, MagPlex) utilizing Luminex xMAP technology and Proseek plates employing Proximity Extension Assay (PEA) technology for biomarker detection.

**Table 3 cancers-15-05607-t003:** Summary of study multiplex technology using advanced immunoassay.

Paper	Biomarker	Target Cohort	SN%	AUC	Ref.
Chen et al. (2018)	CA125, HE4, CEA	Benign, Malignant	88.52	0.972	[64]
Guo et al. (2019)	CA125, MIF, OPN, IL-8 AAb	Early-stage	82	0.974	[72]
Kampan et al. (2020)	CA125, HE4, IL-6, IL-8, IL-6 + CA125, Il-6 + RMI score, IL-6 +HE4, IL6 + ROMA	Benign, Malignant	-	>0.9	[40]
Yang et al. (2020)	CA125 + HE4 Ag-AAb	Early-stage	-	0.986	[73]
	CA125 + HE4 Ag-AAb	Late stage	-	0.985	
Boylan et al. (2017)	CA125, HE4, MK, KLK6, hK11, CXCL13, FR-alpha, IL 6, TNFSF14, FADD, PRSS8, FUR	Early-stage, Healthy	93	0.99	[42]
Leandersson et al. (2020)	CA125, HE4, CXCL6, CTSV, CEACAM1, S100A4, FOLR1,	Benign, Malignant	-	0.921	[39]
	HE4, CA125, ITGAV, CXCL1, CEACAM1, IL-10RB	Benign, Malignant, Borderline	-	0868	
Mukama et al. (2022)	CA125, HE4, KLK11, CXCL13, FOLR1, WISP1, MDK, MSLN, ADAM8	Malignant, Control	-	≥0.70	[74]

Abbreviations: CA125: Cancer Antigen 125; HE4: Human Epididymis protein 4; CEA: Carcinoembryonic Antigen; MIF: Migration Inhibitory Factor;OPN:Osteopontin; IL8: Interleukin 8; IL-8: Interleukin 8 AAb; Anti-IL-8 Autoantibodies; IL-6: Interleukin 6; RMI score:Risk of Malignancy Index; ROMA:Risk of Ovarian Malignancy Algorithm; HE4 Ag-AA: Anti-HE4 autoantibody; MDK: Midkine; KLKs: Kallikreins; hK11: Human kallikrein 11; CXCL13: chemokine (C-X-C motif) ligand 13; FRα: Folate receptor alpha;TNFSF14: tumor necrosis factor superfamily member 14; FADD: Fas-associated protein with death domain; PRSS8: Prostasin; Furin: FUR; CXCL6: hemokine (C-X-C motif) ligand 6; CTSV; cathepsin V, CEACAM1: Carcinoembryonic antigen-related cell adhesion molecule 1; S100A4: S100 calcium-binding protein A4 (S100A4); FOLR1: Folate Receptor Alpha; ITGAV: Integrin alpha V, CXCL1: chemokine (C-X-C motif) ligand 1; IL-10RB:Interleukin 10 receptor subunit beta; FOLR1: Folate Receptor Alpha (FOLR1); WISP1: WNT1-inducible-signaling pathway protein 1; MSLN: Mesothelin; ADAM8; Rabbit Anti-Human ADA;ECLIA Electrochemiluminescence; PEA: Proximity Extension Assay technology.

#### 4.2.1. Luminex Technology

Over time, Luminex such as Luminex xMap INTELLIFLEX, Luminex^®^ 200™, FLEXMAP 3D^®^ and MAGPIX system (Luminex Corporation, Austin, TX, USA) has gained increasing popularity among researchers in comparison to flow cytometers due to its extensive multiplex capabilities and the ability to conjugate limited and complex microbeads with analytes [75,76]. Luminex offers higher SN for detection when compared to ELISA technology using the same antibody pair, typically achieving detection SN at the picogram level. In 2020, Martins et al. further advanced Luminex by incorporating advanced fluidics, optics, and digital signal processing, resulting in the development of multiplexes that utilize fluorescent detection [77]. Luminex’s microbead particles employ fluorescent dyes to generate unique “barcodes” for identification. These particles can be combined to simultaneously analyze multiple target molecules, with the antibodies attached to the microbeads and mixed in a 96-well plate. The microbeads are then analyzed as a single bead suspension through a flow chamber similar to a flow cytometer, and fluorescence intensity is measured [75].

In 2019, Guo et al. compared two different models using early- and late-stage sets of biomarkers. The combination of biomarkers demonstrated a significantly higher AUC of 0.974 (95% CI 0.957–0.991) for early-stage disease compared to an AUC of 0.947 for CA125 alone [74,78]. In the validation set, the addition of OPN, MIF, and anti-IL-8 AAb improved the detection of early-stage cancers from 65% with CA125 alone to 82%. Another study published in 2020 by Kampan et al. distinguished EOC malignancy from benign cases using a 12-plex Luminex assay model that included CA125 and HE4 [40]. The authors found that the combination of CA125 and IL6 yielded a higher AUC compared to CA125 or HE4 alone, with an AUC of 0.643. Yang et al. discovered that 38% of early-stage cases and 31% of late-stage cases exhibited elevated HE4 Ag-AAb complexes, and the combination of HE4 Ag-AAb complexes and CA125 demonstrated a significantly greater AUC (0.986) than CA125 alone (0.879) in early-stage disease (*p* < 0.001) and a slightly greater AUC (*p* = 0.055) [72].

#### 4.2.2. Proximity Extension Assay (PEA) Technology

PEA has established itself as a popular method for homogeneous protein detection, serving as both a biomarker discovery tool and a confirmatory assay. The introduction of multiplexing with PEA technology has brought about a revolution, particularly in the detection of EOC biomarkers, attracting significant attention. PEA has been successfully employed in biomarker discovery for various diseases, including cancer [78].

The PEA platform (Olink Proteomics AB, Uppsala, Sweden), as presented, offers the capability to simultaneously detect 92 proteins in 96 samples, creating possibilities for large-scale generation of protein biomarker data [58]. PEA technology excels in three crucial immunoassay parameters: high SN%, high SP%, and low sample consumption. This technology quantifies multiple biomarkers by employing antibody detection methods that rely on specific detection, coupled with the SN of PCR, resulting in accuracy comparable to other multiplex detection methods [36]. The high-throughput capacity and reproducibility of PEA make it highly desirable for large-scale longitudinal studies, including multi-omics analyses using Proseek plates (e.g., II, Iv2), which yield similar values to those obtained with microbead-based assays. However, the Proseek Oncology Iv2 plate was not specifically designed for EOC.

Boylan et al. utilized a naïve Bayes classifier that combined these 12 biomarkers point the SN% at 95, SP% from 93% to 95%, and the AUC from 0.979 to 0.99 when compared to using CA125 alone [42]. Although the increase in specificity by 2% may seem small, it could have a significant impact on accurately identifying women during large-scale population screening. Leandersson et al. conducted a study using the least absolute shrinkage and selection operator (LASSO) algorithm and identified six biomarkers along with age as the best-performing model for discriminating between benign ovarian tumors and EOC, including borderline tumors [39]. In contrast, the model achieved a diagnostic accuracy with an AUC of 0.868 compared to the reference model (AUC = 0.770) consisting of HE4, CA125, and age (*p* = 0.016 at best point cut-off). However, another model successfully discriminated between benign tumors and EOC with an AUC of 0.921 at the best point cut-off (*p* = 0.025) [39]. Similarly, Mukama et al., used the LASSO algorithm on eight combination biomarkers with CA125 did not achieve AUC ≥ 0.70, addressing the discriminatory power of these biomarkers was lower when the sample collection lag-time was more than 9–18 months prior to diagnosis [74,79]. The theoretical findings of this multiplex biomarker panel indicated insufficient improvements in model fit or diagnostic performance.

### 4.3. Challenges of Multiplex Technology Using Advance Immunoassay

#### 4.3.1. Target Cohort and Sample Size

The challenges in EOC biomarker research include study population heterogeneity, limited sample size, and the lack of a tailored multiplex assay. These issues significantly impact the interpretation of statistical analyses and hinder the accurate diagnosis of EOC within specific target populations. This variation in patient characteristics makes it difficult to draw conclusive results and generalize the findings to larger populations. Additionally, the use of unicentric approaches limits the diversity and representation of the study cohort, further undermining the reliability of the results. Many published studies on EOC biomarkers suffer from small variations in sample sizes, which compromises the generalizability of their findings [80]. Working with small sample sizes may result in a reduced matrix effect, where the sample composition and surrounding factors have a lesser impact on the EOC biomarker panel’s performance [36,70]. Therefore, to accurately assess the effectiveness of a biomarker panel in detecting early and late stages of EOC, larger and more diverse sample sizes are required [61].

#### 4.3.2. Dependency-Appropriate Biomarker Combinations

The combination of biomarkers in an EOC multiplex panel holds promise for enhancing the accuracy of EOC detection. When additional biomarkers exhibit a negative correlation with primary biomarkers, they can contribute to the predictive power of the panel [4]. However, the inclusion of extra biomarkers does not always result in a significant improvement in accuracy, particularly if they are highly correlated with the primary biomarkers [45,62]. For instance, well-studied biomarkers like CA125 and HE4 may not substantially enhance the AUC of the panel if they are strongly correlated with the primary biomarker [36]. Therefore, in order to achieve high prediction accuracy, it is crucial to carefully select biomarkers and understand their associations within the EOC panel. Extensive research and analysis are necessary to identify biomarkers that demonstrate the required associations and offer the best prediction accuracy. This involves investigating the relationships between primary and secondary EOC biomarkers, as well as evaluating their individual and combined effects.

#### 4.3.3. The Technologies

The utilization of multiplexing technology, such as advanced Luminex and PEA immunoassay, has become crucial in the detection of EOC biomarkers. These technologies offer several advantages over conventional detection methods and have the capability to achieve a high level of accuracy, with an AUC greater than 0.9. Luminex technology utilizes fluorescent techniques and flow cytometer separation with different dyes or bead sizes, enabling accurate detection [81,82]. However, Luminex technology faces challenges in detecting low molecular weight (*M_W_*) biomarkers in small sample volumes. Guo et al. have found that the detection of HE4 using Luminex was elevated in only 12% of early-stage EOC cases and 36% of late-stage cases [72]. Furthermore, labeled multiplex immunoassay formats may struggle to accurately identify complex antibodies and differentiate between different antibody complexes [57,83,84]. This is attributed to the inconsistent affinity and stability of target analytes in multiplex assays that utilize monoclonal antibodies [76,85].

On the other hand, advanced multiplex immunoassays, such as PEA, allow for the water simultaneous detection of multiple biomarker combinations and offer highly sensitive detection of target proteins through the amplification of DNA templates (As referenced in Table 4, the study provides a summary of the comparison). However, PEA is not widely used in clinical applications compared to Luminex. It is important to note that both Luminex multiplex microbeads and proximity-dependent amplification assays have certain limitations in detecting analytes when sandwiched between specific antibodies, as well as in identifying complex antibodies, which can be expensive [57,84,86].

#### 4.3.4. The EOC Multiplex Signatures

Achieving the optimal balance between S_N_ and S_P_ poses challenges for the EOC multiplex signatures. While PEA technology exhibits high S_N_ and S_P_, there can be a trade-off in specificity, leading to reduced overall accuracy [79,87]. Therefore, careful selection of the immunoassay platform is crucial to ensure reliable and accurate results. Currently, Luminex multiplex biomarkers lack the necessary accuracy to specifically identify early-stage EOC with high specificity, resulting in false positives and low specificity rates. Enhancing the accuracy and addressing batch variation in the multiplex panel is essential to reduce false positives and improve specificity.

Standardization and validation of EOC biomarker signatures are necessary to ensure consistent and reliable interpretation of data. While meta-analyses and public databases have been useful, further research involving large-scale clinical studies and the development of a comprehensive database is required. Extensive cross-validation, both at a technological level and in interdisciplinary clinical trials, is crucial to achieving validation of the multiplex technology for EOC biomarkers [86]. Rigorous validation of each EOC biomarker within the panel is necessary, assessing various performance characteristics such as sensitivity, specificity, accuracy, precision, linearity, and dynamic range [79].

Reducing false positives is of utmost importance to alleviate anxiety, minimize costs associated with the multiplex panel, and avoid potential complications. Therefore, prioritizing extremely high specificity in an EOC screening strategy is essential. The use of signature or pattern results can provide insights into potential mechanisms of action and guide future biomarker research. However, the high cost of these technologies currently limits the number of assays that can be performed in each study.

**Table 4 cancers-15-05607-t004:** A comparison of Luminex and PEA technology, highlighting their respective limitations [36,70,81,82,83,88,89].

	Multiplex Technology	Luminex Technology	PEA Technology
Key Points			
Principle		Bead-based Immunoassay	Proximity ligation and Amplification
Number of analytes		100	92
Sensitivity		Variable, Depending on the assay design, The dilution factor of an analyte	High sensitivity
Application		Protein biomarker analysis, cytokine profiling	Protein biomarker analysis, New biomarker discovery, Drug development
Commercial availability		Widely	Available from specific vendors
Limitation		Expensive biomarker panel/assay Complexity and technical expertise Cross-reactivity and specificity Potential interference from the sample matrix Limited customization panel Requires extensive validation	

## 5. Point-of-Care (POC) Multiplex Sensors for Detecting EOC Biomarkers

Recently advanced biosensor technology has played a significant role in correlating cancer biomarker profiling data, particularly in multiplexing POC devices. These devices offer improved LOD ranging from picomolar to femtomolar, ensuring accurate and precise results. Another area of active interest has been the development of microfluidic/lab-on-a-chip (LOC)-based approaches that offer real-time POC results to patients [79,87]. In previous work, SPR biosensor has been used to detect prostate-specific antigen by using a multi-channel LoC system that combined micropatterned plasmonic materials with SPR, fluorescence detection, and microfludics [90,91,92,93].

The specifications of POC biosensor devices are crucial for integrating all aspects of an assay onto a handheld platform. The development of a multiplexed biosensor for clinical use is essential for identifying and quantifying biomarkers in therapeutic and diagnostic applications. Key attributes for miniaturized biosensors utilizing microfluidics for POC application design include low detection limits and quick analysis LOD, specificity, sensitivity, reproducibility, reliability, robustness, cost, speed, and multiplexing capability [85]. Detecting circulating protein biomarkers in small serum volumes using label-based biosensor systems is challenging due to limited S*_SPR_*. However, current immunoassay multiplex biosensors offer improved LOD, ranging from picomolar to femtomolar, to ensure accurate and precise results [90,94,95,96].

Figure 2 shows the illustration of a typical biosensor which consists of five main elements: analytes (substances to be detected), bioreceptors (specific molecules that interact with the analytes), transducers (electronics that amplify and detect signals), digital circuitry (amplifier and processor), and an interpretation system (displays the output) [27,81,97]. This enables highly sensitive detection in small volumes. Biosensors use various physicochemical transducers like optical, electrochemical, thermal, and piezoelectric, which are biocompatible, have a high surface-to-volume ratio, and maintain bioreceptor activity [21,81]. Biosensors can be classified based on the combination of bioreceptor-analyte, detection systems, and technologies. Multiplexed biosensors combine multiple detection elements to enhance sensitivity and precision for detecting EOC biomarkers, sometimes using bio-recognition elements such as nanoparticles (NP), molecularly imprinted polymer (MIP), and others.

### Label-Free Optical Biosensor

Over the years, label-free biosensors have rapidly advanced to detect low molecular weight biomarkers on POC platforms with minimal input and real-time detection capabilities [28,84]. Optical transducers, which use various physical and chemical principles to detect changes in sensor properties caused by target molecule binding, have gained popularity [85]. This method enables real-time monitoring of binding reactions and provides access to kinetic and thermodynamic parameters [26,85,98].

In contrast, label-based sensing methods generate optical signals through calorimetric, fluorescence, or luminescent techniques. Optical biosensors can be designed based on various principles, including SPR evanescent wave fluorescence, optical waveguide interferometry, chemiluminescence, fluorescence, refractive index (RI), and surface-enhanced Raman scattering (SERS) [27]. Optical immunoassays offer multiplexed detection and are attractive for their ability to recognize and bind specific target analytes. However, achieving high sensitivity and detection limits requires optimization and validation processes.

## 6. Surface Plasmon Resonance (SPR) Optical Biosensing

SPR is a widely used optical biosensor that operates without the need for labeling. It allows for the real-time detection of biomolecules and offers great versatility compared to other optical biosensors [25]. SPR enables the quantitative and real-time detection of individual biomolecules and is well-suited for high-throughput analysis [87,99].

The SPR principle relies on alterations in the RI at the surface plasmon interface, which occur due to the binding of molecules within the detection region, as depicted in Figure 3 [88,99]. When biomolecules bind to the metal-dielectric interface, they create an ultra-thin organic layer on the metal, thus modifying the resonance condition for surface plasmons [43]. Figure 3B illustrates the SPR angle shift (from I to II in the lower left-hand side), and Figure 3C presents a sensorgram showing the changes in the SPR signal, which monitors the sample response level during analyte-ligand interactions [25]. The SPR experiment consists of three main phases: a buffer injection phase to establish a baseline signal on the surface, an analyte injection phase where analyte-ligand complexes are formed (association), and a second buffer injection phase to dissociate the analyte-ligand complexes (dissociation) [90,100]. Finally, a regeneration solution is introduced to break the particular connection between the analyte and the ligand [100,101]. The quantification of molecular interactions occurring on the metal layer is expressed in refractive index units (RIU) [102].

In a clinical investigation, SPR should be fully developed after evaluating their performance with analytical parameters, such as dynamic range, S*_SPR_*, selectivity, reproducibility, and stability before they can be optimized for commercial and clinical uses [27,103,104]. However, current studies of newly developed SPR biosensors aim to obtain low LOD and a wider dynamic range. Validation of the SPR analysis can be done by identifying a known marker in serum using a conventional approach such as ELISA and comparing the results obtained from the SPR analysis to ensure accuracy and reliability [2,28,102,105,106].

In addition to SPR biosensors, other methods such as localized SPR (LSPR) and SPR imaging (SPRi) are being used in clinical trials for detecting EOC biomarkers. Different types of SPR-based biosensors have their own advantages, limitations, and characteristics, which are detailed in Table 5. The LSPR and SPRi methods differ in their imaging and processing techniques upon the immobilization of analytes on the sensing surface. The choice of SPR type depends on various factors for multiplexed detection through various strategies, such as spatial multiplexing, temporal multiplexing, sandwich assay, and competitive assay. Reflectivity-based SPRi provides spatial information about binding events using a wider range of incident angles [96]. This enables multiplexed sensing detection and real-time visualization of biochip surfaces without interferences of non-specific binding [96]. Additionally, the use of LSPR has emerged as a fabrication strategy for highly sensitive and specific biomarker assays. Utilizing NPs on a metal film offers the advantages of multiplexing and high throughput capabilities [99].

There are several more methods for generating SPR with trendily utilized fabrication strategy, such as creating Molecularly Imprinted Polymers (MIP), including the formation of the 3-D polymeric matrix, making nano-sized scaled sensor surfaces or creating a rough surface using NPs or nanomaterials (NMs), nanoplate and nanowires on metal films such as gold (Au) or silver (Ag) [109,110]. Interestingly, both metal films (Au and Ag) are coated on a glass which provides strong plasmonic properties [2,99]. Most commonly, Au is often preferred over Ag because it is more stable and less likely to undergo oxidation or other chemical reactions that could affect the SPR signal [2,102,104,111,112]. It also has the strongest response at the higher wavelengths. However, Ag has a higher S*_SPR_* to changes in RI than Au, making it useful for detecting smaller molecules in sufficient sample concentrations and it is also cost-effective [113,114].

Whereby, the combined usage of various NMs on metal films (such as Au NP and Ag NP), magnetic NP (like quantum dots and carbon nanotubes), and nanomaterials such as photonic crystals (PhC), and graphene materials offer excellent biocompatibility, unique conductivity, and optical properties in SPR platforms [87,99,105,115,116,117]. Moreover, metal films have been employed to enhance the local electromagnetic field and increase the efficiency of Raman scattering by several orders of magnitude, leading to SERS [118,119].

### 6.1. SPR in Multiplexed Detection of EOC Biomarkers

Interestingly, a few studies investigated combinations of two or three biomarkers for EOC using different transducers for biosensors. In 2022, Wu et al. developed a fluorescence-based biosensor utilizing nanomaterials to simultaneously detect CA125, HE4, CEA, and AFP biomarkers [120,121]. This biosensor achieved impressive results, with a low LOD of 0.1 U/mL for CA125, approximately 10 pg/mL for HE4, and 1 pg/mL for CEA, and AFP. Although these findings demonstrate the potential of multiplex biosensors, it is essential to consider that the biomarker concentrations may vary depending on the measurement method, particularly when employing a SPR benchtop version. Consequently, prioritizing the study of SPR biosensors in multiplex formats would be crucial to further enhance the detection limits of EOC biomarkers.

Currently, no existing study has developed a multiplex SPR biosensor specifically for diagnosing EOC using human serum. Recent SPR-based biosensors designed for multiplex EOC biomarker detection, offering increased S*_SPR_*, throughput, and features for fragment-based work, have emerged in the market. However, there is a lack of detailed protocols or descriptions of generally suitable experimental procedures and disclosed methods for more challenging targets remain limited.

Nonetheless, this study underscores the potential of utilizing SPR biosensors for the simultaneous detection of CA125 and HE4 biomarkers in EOC. The objective is to identify an SPR method capable of effectively detecting low concentrations of various combinations of EOC biomarkers, including CA125 and HE4. The development of a multiplex SPR biosensor would hold significant value in healthcare, particularly for monitoring EOC biomarkers. These findings are crucial in guiding the selection of appropriate SPR instruments and immobilization techniques to construct a tailored multiplex platform for specific clinical applications.

### 6.2. Detection of CA125 and HE4 Biomarkers Using SPR

This section aims mainly to summarise the performance of SPR in the detection of CA125 and HE4 biomarkers, selected over the past 5 years. Table 6, shows the detection of CA125 and HE4 using SPR techniques, highlighting the LOD and dynamic range. Table 7 shows the evaluation of the immobilization technique, key features as well and signal amplification in terms of advantages and limitations.

#### 6.2.1. SPR-Based Biosensors

Recent research has shown that the MIP technique has promising results in clinical cancer diagnosis which was initially proposed by Pauling et al. for synthesizing artificial antibodies using antigens as templates in a polymer matrix. MIP has been used on Au nano-electrode ensembles and polyphenols as a monomer in the detection of CA-125 [28]. Rebelo et al. developed a three-dimensional MIP based on SPR in a thin film to detect CA125 by combining square wave voltammetry (SWV) and SPR sensors [111]. The detection of CA125 in artifical serum was observed to have a wide concentration range of 0.1–500 U/mL and a lower LOD of 0.1 U/mL, where a gold screen-printed electrode (Au-SPE) was developed with MIP technique and on an SPR gold sensor with monomer pyrrole (Py). The recovery rates for CA125 in artificial serum samples ranged from 91% to 105% according to the kinetic results.

Lastly, Yi et al. conducted a theoretical study using the sandwich immunoassay method, which found that AuAg-SPR sensors had higher *S_SPR_* than conventional Au-SPR sensors for detecting CA125 [113]. They observed changes in the resonance wavelength and a low LOD when using AuAg sensors, demonstrating direct detection of CA125 at a concentration of 0.1 U/mL (0.8 ng/mL). In contrast, the Au sensor showed less S*_SPR_* in their direct reaction with an immobilized Ab, with no change in the resonance wavelength (λR) for two concentrations of the CA125 solutions. Additionally, AuAg has a good correlativity of 0.00948 compared to the Au sensor.

#### 6.2.2. SPRi

Gur et al. designed SPR chip surfaces that were coated with CA-125 which was imprinted and non-imprinted with poly (2-hydroxyethyl methacrylate-N-methacryloyl-(L)-tryptophan methyl ester) nanoparticles (NP) [122]. The CA125 detection response was a newly developed technique that discovered ultrasensitive LOD between 0.01 U/mL, which was lower than the dynamic range of 0.1–10 U/mL [109,122]. They demonstrated their method with an artificial serum that has been very effective in the detection of CA125 levels in complex media. On a nanoscale chip that could be maintained for five months, the interaction between CA-125 imprinted SPR sensors and the CA-125 biomarker was examined five times without background interference.

Szymanska et al. constructed a SPRi for the determination of CA-125 using a thiol-modified gold surface with a thickness of 50 nm with a cysteamine (CysA) linker bonded to the gold surface. The linker contains a mercapto group (compounds containing sulfur) that consists of numerous carboxyl groups capable of reacting with the amine groups of CysA, as it is suitable for detecting CA125 biomarker within a 2.2 (LOD)-50 U/mL^−1^ dynamic range [97]. Comparison studies between SPRi and ECLIA between EOC patients show significant differences. CA125 recovery is satisfactory at 1.04 U/mL, which was performed on the edge of the range of applicability of the method.

Additionally, Szymanska et al. demonstrated HE4 biomarker detection using a non-fluidic array SPRi based on a single Ab using a cysteamine linker covalently linked to a gold chip and immobilized rabbit polyclonal antibody against HE4 (114 pM) [43]. According to Yuan et al. studies, an LSPR-based HE4 biosensor was developed using an anti-HE4 antibody mounted on a nano silver chip through an 11-mercaptoudecanoic linker. Szymanska et al. technique detected lower dynamic detection at 2 pM than Yuan et al. result [28,123]. They found that the SPRi immunosensor caused the dilution of more concentrated samples to eliminate non-specific binding. In addition, HE4 concentration in a patient pre- and post-surgery was stable with the immobilized HE4 concentration in serum dramatically reduced (Mw of 44 kDa) 5 days after tumor resection, showing that this method is successful in the clinical setting.

#### 6.2.3. LSPR Co-Enhanced Raman Scattering

Recent studies have provided valuable insights into the relationship between Localized Surface Plasmon Resonance (LSPR) and Co-Enhanced Raman Scattering, resulting in the generation of detailed spectral data through enhanced plasmonic effects [124]. LSPR waves are produced in conventional Raman spectroscopy, which was designed for the nanoscale level, at interstitial or sharp edge places on the substrate [114]. This improved analytical performance and organic insights may be gained by combining Raman spectroscopy with SPR approaches, leading to a more complete understanding of surface interactions and processes [115].

Raman spectroscopy is a versatile, label-free method with extensive use in cancer diagnoses and material analysis [119]. This technique has been extensively used in cancer diagnosis and analysis [116]. While Raman scattering is known for its sensitivity, its utility is constrained by poor signal strength. This limitation has spurred the development of amplification methods such as Surface-enhanced Raman spectroscopy (SERS) [124]. SERS utilizes rough metallic surfaces or nanostructures to significantly amplify the Raman signal by orders of magnitude 10^3^–10^10^ M times. Eom et al. found strong SERS immunoassay signals as it allowed HE4 to be detected at low concentrations in attomolar of 10^–17^ M using Au nanoplate (NPl) [114]. However, the dynamic range was 10^−17^ to 10^−9^ M when the HE4 concentration was increased.

Signal intensity in LSPR-SERS is greatly amplified by the distribution of the electric field in the hot-spot region, leading to a significant increase in intensity. Researchers have conducted numerous studies that focused solely on the development of SERS-active substrates using Au and Ag NPs [118]. Ryu et al. developed SERS intensity as a result of the strong coupling between the SPR of the Ag crater structure, the LSPR of the Ag metal film, and the LSPR among the Ag nanowires (NWs) but these interactions significantly influence the LSPR-dependent performance of SERS [110]. These studies have shed light on how the wavelength of LSPR impacts the enhancement factors and the mapping of Raman signal intensities over time, as well as the S_SPR_ of SERS measurements.

LSPR band response shifting through AuNPs film gave hot-spot building, which has greatly increased electromagnetic fields of the regional distribution of SERS amplification on the CA125 at nM concentration level. CA125 Ab-Ag bioconjugation was performed with AuNPs resulting in a changed response in the Raman spectrum red-shifting in the LSPR band, which was relatively minimal at 200–220 nM concentration as recorded by TunC et al. [120].

### 6.3. Challenges and Future Preceptive in Developing New SPR-Based Biosensor

It is crucial to validate newly developed SPR biosensors for the evaluation of EOC biomarkers, and close collaboration between analysts and clinicians is recommended. Access to clinically characterized serum samples is unrealistic without such cooperation. Besides detecting biomarkers, there is a need to develop SPR-based methods for determining the stage of EOC or monitoring cancer therapy progress. The study on SPR detection of CA125 and HE4 biomarkers showed a lack of standard methods and key features for signal amplification on SPR and SPRi. Challenges still exist, such as the duration of the antigen of interest, long-term stability of bio-recognition molecules, stable interface potentials, suitable inhibitors for immobilized biomarkers, and selection of appropriate surfaces or matrices for their expression.

Establishing newly developed SPR for detecting low concentrations of EOC biomarkers in complex matrices requires techniques such as MIP and NP. MIP-based sensors have shown promise but can be expensive and require sophisticated hardware for clinical use. The success of MIP-based sensors depends on factors like protein attachment, polymerization techniques, and monomer selection. In 2020, Gur et al. also demonstrated a low LOD of 0.01 U/mL for CA125 using a modified MIT technique. These findings suggest that SPR with the MIT technique may be suitable for constructing a multiplex platform to detect EOC biomarkers [122].

Additionally, the stability of the sensor chip and modified metal sensors developed with SPR biosensors was compared, and Gur et al. found that only the chip with MIT and a nanoscale chip maintained stability for 5 months, while the cysteamine layer remained stable for one month. In addition, the conventional use of a single thin layer of gold in SPR biosensors may not meet the S*_SPR_* requirements. However, the multi-layer/material SPR such as Ag/Au chip and AuNP in SPR, demonstrated sharper response characteristics and lower manufacturing costs compared to an Au chip [28]. Flexible biomarker systems and enzyme-mediated detection are still prevalent in the SPR detection of EOC biomarkers.

Challenges in sampling include determining minimum sample volumes and replenishment rates. Delayed reactions to antigens of interest may also arise due to the isolation of circulating proteins from serum. Overcoming challenges in developing selective SPR biosensors for EOC diagnosis involves identifying the appropriate immobilizer, suitable matrix, and optimal Ag/Au chip for enhanced S*_SPR_*. Table 5 includes a classification of SPR-based biosensors, highlighting their advantages and disadvantages for different applications and analytes. The choice of biosensor technique, whether SPR, LSPR, or SPRi, depends on the specific application and experimental requirements. Validation formats should be established to ensure material and application suitability in future advancements of SPR sensors.

Szymanska et al. conducted clinical trials using an SPRi-based chip to monitor the immobilization of HE4 before and after surgery, comparing the results with the standard ECL method. Although some of the developed SPR biosensors have been partially characterized in terms of analytical parameters, further validation is needed, which can be addressed in future commercial evaluations. SPR-based biosensors have garnered significant attention in evaluating the performance of EOC biomarkers, particularly in terms of LOD, dynamic range, and stability. Available data suggests the potential of SPR in quantifying EOC biomarkers at lower concentrations in real biological samples, although its full implementation in clinical diagnosis is still being realized.

Some challenges are related to the wavelength used in the detection process and the selection of the optimal S*_SPR_* through the choice of wavelength. The SPR resonance wavelength of light and the dielectric properties of the surrounding medium depend upon the RI of the detecting biomarkers [125]. Few SPR research papers were successfully used to detect various concentrations of glucose [93], and simulation data from numerical modelling was compared with the results achieved at optical wavelengths of 670 nm and 785 nm [126].

The utilization of specific wavelengths in the detection process and the careful selection of the optimal S*_SPR_* poses certain challenges in SPR technology. The resonance wavelength of light and the dielectric properties of the surrounding medium are dependent on the RI of the detecting biomarkers [125]. Table 6, provides a list of commonly employed SPR wavelengths for detecting CA125 and HE4, typically falling within the range of 600 nm to 700 nm, except for the case of 532 nm used by Tun C et al. [120]. Notably, a few SPR research papers achieved successful detection of various glucose concentrations [93] and another study comparing different sensing layers comprising multiple layers of graphene results obtained at optical wavelengths of 670 nm and 785 nm.

On the other hand, the combination of Raman spectroscopy and SPR techniques enhances analytical performance and provides valuable chemical insights, improving our understanding of surface interactions and processes [110]. While SERS overcomes limitations in traditional Raman spectroscopy such as signal intensity and specificity, it has its own drawbacks including reproducibility issues and the potential impact of metal-molecule binding on result consistency and reliability [114,116,119]. In a study by Tun C et al., CA125 detection was achieved using LSPR-SERS, but the interaction between Ag nanoparticles and the antigen was hindered due to the antigen’s distant position from the antibody [120]. These limits are indicated in a higher standard deviation variation among class means, which highlights the need for precise optimization when using SERS in different applications.

## 7. Conclusions and Outlook

The growing demand for validated multiplexing, which detects multiple analytes simultaneously, improves biomarker detection in serum. Different approaches have been used, including labeled-based assays and technologies like Luminex multiplex and PEA. However, these methods lack the dynamic range and robustness of newer paramagnetic and fluorescent assays. Therefore, the challenge is to customize a panel of biomarkers for EOC detection with higher accuracy and specificity. Establishing standards for accuracy, precision, and reproducibility is crucial for analyzing large datasets from multiplex panels. While multiplexing is desired in research settings, it is often inaccessible due to the high instrumentation costs. Current technologies have not met critical performance parameters such as specificity, sensitivity, LOD, and dynamic range. This article emphasizes the critical need for validation of potential biomarkers and new technologies that are discussed in exploratory studies. While these exploratory studies provide valuable insights and promising findings, it is crucial to validate their effectiveness and reliability in future clinical studies.

These principles can enhance assay sensitivity, expand the dynamic range without sample dilution, and decrease incubation time in research laboratories. Implementing multiplexing in POC devices for EOC biomarker detection offers advantages such as simultaneous detection, improved accuracy, faster results, simplified quantification, cost savings, and reduced sample volume requirement. However, the main challenges in the research and development of surface plasmon resonance (SPR) sensors are the expensive platforms and components. Typically, commercial platforms are not affordable for small research groups or POC settings to invest in. Hence, this study holds potential as a roadmap for scientists and engineers in academia and industry to develop a low-cost yet highly effective SPR sensor platform.

This review examines the utilization of SPR and SPRi in the past five years for the detection of CA125 and HE4 biomarkers. It discusses the operational principles and applications of specific SPR, LSPR, and SPRi devices in selectively detecting various tumor markers. The evaluated biosensors demonstrate low limit of detection (LOD) values for cancer biomarker detection in different sample sources. The results indicate notable progress in SPR detection for clinical diagnosis. Label-free detection methods like SPR show great promise, achieving LOD values of 0.01 U/mL^−1^ for CA125 and 1 pM for HE4, enabling real-time detection. However, studies from 2019 to 2022 still face challenges in optimizing method selection, materials, and signal amplification, and have limitations in accuracy compared to standard immunoassay techniques. Several commercially available SPR instruments, including Biacore™ T200 (Cytiva, Uppsala, Sweden) and MP-SPR Navi™ (BioNavis, Tampere, Finland), facilitate detection. The integration of SPR with modified sensor surfaces using MIP and nanomaterials, combined with multiplexed biosensors, holds significant potential for detecting EOC biomarkers. The emergence of SPR sensors offers the potential to enhance the use of post-operative biomarkers for clinical prognosis and cancer treatments, thereby reducing expenses and improving healthcare outcomes.

## Figures and Tables

**Figure 1 cancers-15-05607-f001:**
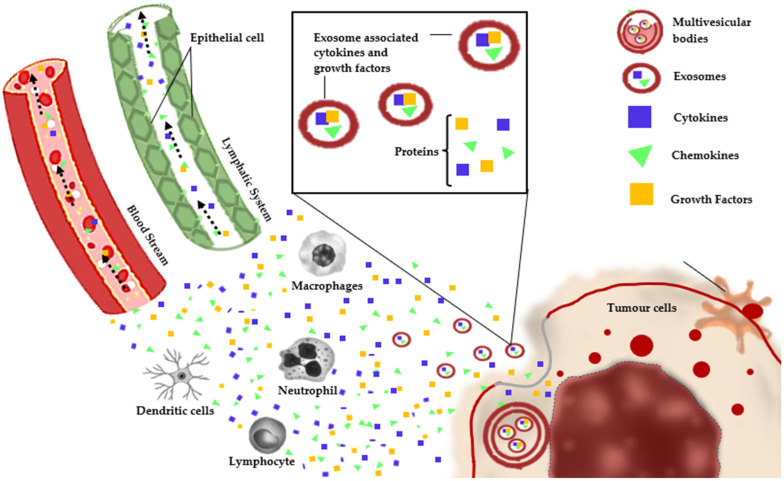
Schematic illustration of interactions between tumor cells and immune cells in TME components and exosomes [4,6,25,28,29].

**Figure 2 cancers-15-05607-f002:**
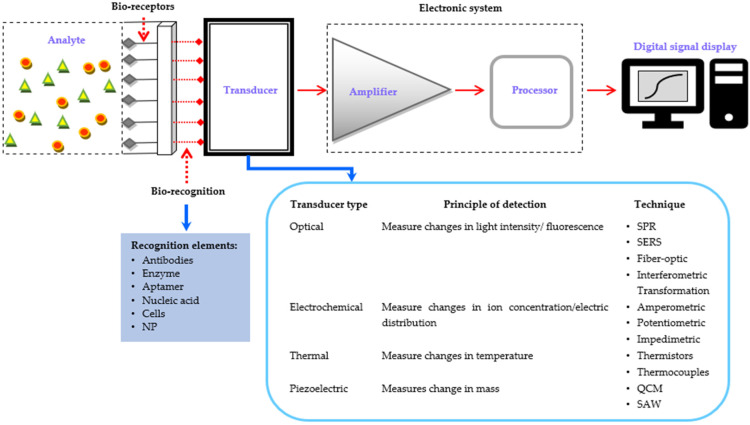
A basic illustration of biosensors comprising of analytes, bioreceptors, bio-recognition elements, transducers with their variation, electronic system, and digital signal display [21,27,81,97].

**Figure 3 cancers-15-05607-f003:**
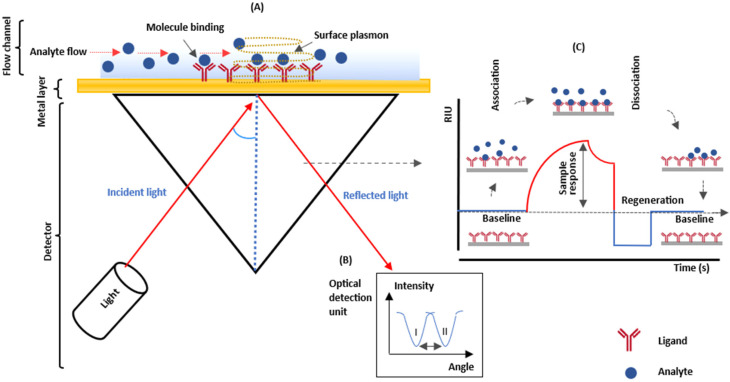
(**A**) Schematic illustration of SPR system. (**B**) The SPR angle shift. (**C**) Sensorgram a showing the steps of an analytical cycle [25,88,90,99,100,101,102].

**Table 5 cancers-15-05607-t005:** List of different SPR-based biosensor, their advantage and limitations, and their characteristics.

Biosensor Types	Advantages	Disadvantages	Type of Signal	Surface Substrate	Ref.
SPR	High S*_SPR_* Monitoring binding events in real-time	Non-target binding Limited multiplexing and measuring capabilities.	RI	Metal sensor thickness	[2,81,107]
LSPR	High Specificity Utilization of Gold NP	Surface immobilization based on structural modifications. Complex mechanism of action	Absorption peak shift	Particle size and shape	[28,81,108]
SPRi	Higher reflection Spatial multiplexing	Non-specific interferences Complicated surface functionalization process	Intensity contrast image	Control the intensity of light with a fixed angle	[96,99]

Abbreviations: SPR: Surface plasmon resonance; SSPR Sensitivity; LSPR: localized SPR; SPRi: SPR imaging; NP: nanoparticles; RI: Refractive index.

**Table 6 cancers-15-05607-t006:** Represents the list of SPR used to detect CA125 and HE4 with LOD and dynamic range.

Platform	Biomarker	LOD	Dynamic Range	Wavelength (nm)	Ref.
SPR	CA125	0.1 U/mL^−1^	0.1–300 U/mL^−1^	670	[111]
		0.1 U/mL^−1^	0.1–1 U/mL^−1^	600–700	[113]
SPRi	CA125	0.01 U/mL^−1^	0.1–10 U/mL^−1^	670	[122]
		2.2 U/mL^−1^	2.2–150 U/mL^−1^	633	[97]
	HE4	2pM	2–110 pM	-	[43]
LSPR-SERS	CA125	20 ug/mL	-	532	[120]
	HE4	10^−17^ M	10^−17^–10^−9^ M	633	[114]

**Table 7 cancers-15-05607-t007:** Evaluation of immobilization technique, key features, and signal amplification in terms of advantages and limitations.

Paper	Recognition Element	Metal Flim/Fabrication Strategy	Stability	Advantages	Limitation
Rebelo et al. (2019) [111]	MIP	Au/Py/SPE	-	Good selectivity. Reusability.	S*_SPR_* to pH, and temperature, which can affect its conductivity and stability
Gür et al. (2020) [122]	Imprinted polymeric NPs	Au/Poly(HEMA-MATrp)/NS	5 months	High stability. Reusability.	Susceptibility to hydrolysis, which can lead to its degradation over time
Szymańska et al. (2020) [97] & (2021) [43]	pcAb	Au-CysA	-	Good recovery. High specificity.	Low absorptivity. Instability.
Yi et al. (2020) [113]	pcAb	AuAg	-	Excellent biocompatibility.	Instability.
TunC et al. (2020) [120]	mAbs/Ag	AuNPs/SERS	-	SERS signal remains consistent even at low concentrations.	Limit the extent of Ag-NP interactions
Eom et al. (2021) [114]	mAbs	Au/NPl/(Cys)3-protein G/SERS	-	High specificity.	Complexity of the substrate preparation.

Abbreviations: MIP: Molecularly imprinted polymer; pcAb: Rabbit polyclonal antibody; mAbs: Monoclonal antibodies; Ag: Antigen; Py: Pyrrole; SPE: Screen-Printed Electrode; Poly (3 HEMA-MATrp): Poly (2-hydroxyethyl methacrylate-N-methacryloyl-(L)-tryptophan methyles; NS: Nanoscale; Poly (3-HPA): Poly 3-hydroxyphenylacetic acid; SPCE: screen-printed carbon electrodes; CysA: Cysteamine; NP: Nanoparticles; (Cys)3-protein G: Cysteine-tagged protein G; NPI: Nanoplate; SERS: Surface-enhanced Raman scattering.

## Data Availability

No new data were created or analyzed in this study. Data sharing is not applicable to this article.

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
