# Peer review of "Recent Advances in Surface Plasmon Resonance (SPR) Technology for Detecting Ovarian Cancer Biomarkers"

_cancers, 2023, doi:10.3390/cancers15235607_

Round 1

Reviewer 1 Report

Comments and Suggestions for Authors

Dear Authors,

Cancers-2693279:

Recent Advances in Surface Plasmon Resonance (SPR) Technology for Detecting Ovarian Cancer Biomarkers by Ravi Kumar etal is an interesting review article. This review article proposes a potential roadmap for scientists and engineers in academia and industry to develop a cost-effective yet highly efficient SPR biosensor platform for detecting EOC.

Positive Comments:

1.     This review examines the utilization of SPR and SPRi in the past five years for the 748 detections of CA125 and HE4 biomarkers.

2.     This review discusses the operational principles and ap- 749 plications of specific SPR, LSPR, and SPRi devices in selectively detecting various tumor 750 markers. The evaluated biosensors demonstrate low limit of detection (LOD) values for 751 cancer biomarker detection in different sample sources.

3.     Interestingly label-free detection methods like SPR show 753 great promises, achieving LOD values of 0.01 U/mL-1 for CA125 and 1pM for HE4, ena- 754 bling real-time detection.

Comments on the Quality of English Language

Minor Edits needed

Author Response

We would like to thank the Editor and Reviewers for their constructive comments and suggestions. Highlighted in yellow are our responses to the queries and correction made in the manuscript.

REVIEWER #1 COMMENTS AND AUTHOR’S RESPONSES

Minor editing of English language required.

Answer: The Manuscript has been proofread by native English-speaking personnel.

Dear Authors,

Recent Advances in Surface Plasmon Resonance (SPR) Technology for Detecting Ovarian Cancer Biomarkers by Ravi Kumar et al. is an interesting review article. This review article proposes a potential roadmap for scientists and engineers in academia and industry to develop a cost-effective yet highly efficient SPR biosensor platform for detecting EOC.

Positive Comments:

  1. This review examines the utilization of SPR and SPRi in the past five years for the 748 detections of CA125 and HE4 biomarkers.
  2. This review discusses the operational principles and ap- 749 plications of specific SPR, LSPR, and SPRi devices in selectively detecting various tumor 750 markers. The evaluated biosensors demonstrate low limit of detection (LOD) values for 751 cancer biomarker detection in different sample sources.
  3. Interestingly label-free detection methods like SPR show 753 great promises, achieving LOD values of 0.01 U/mL-1 for CA125 and 1pM for HE4, ena- 754 bling real-time detection.

Answer: Thank you for the positive comments.

CHANGES WITHOUT MENTIONED IN COMMENTS

  1. Deleted/RemovedWe have removed the following lines: 43-45 and 45-46, as part of the improved writing regarding epidemiology and therapeutic approaches.

It is often referred to as a "silent killer" because it frequently goes unnoticed until it has already progressed aggressively from an early to an advanced stage, typically within one year.

line 43-45

The majority of EOC patients, more than 70%, receive a diagnosis when the cancer has already reached an advanced stage[3, 5]

line 45-46

  1. Amend at Simple Summary: line12-13

We have improvised the “simple summary” in line 12-13.

Simple Summary: This review significantly contributes to the research community by providing insights into the role of serum-based biomarkers for the diagno

Reviewer 2 Report

Comments and Suggestions for Authors

The authors present a comprehensive overview of recent advancements in immunoassay detection methods, highlighting the use of multiplex technology and SPR biosensors for the identification of biomarkers CA124 and HE4. Additionally, the review also highlights the challenges associated with these diagnostic methods. The study has the potential to contribute significantly to the development of a cost-effective yet highly efficient SPR biosensor platform for detecting EOC. The review is well organized and written on SPR for EOC, providing more useful information for detecting EOC and bringing benefits to EOC patients. It is, therefore, acceptable.

Comments on the Quality of English Language

The review is well organized and written.

Author Response

 We would like to thank the Editor and Reviewers for their constructive comments and suggestions. Highlighted in yellow are our responses to the queries and corrections made in the manuscript.

REVIEWER #2 COMMENTS AND AUTHOR’S RESPONSES

Minor editing of English language required

Answer: The Manuscript has been proof-read by a native English-speaking personnel.

The authors present a comprehensive overview of recent advancements in immunoassay detection methods, highlighting the use of multiplex technology and SPR biosensors for the identification of biomarkers CA124 and HE4. Additionally, the review also highlights the challenges associated with these diagnostic methods. The study has the potential to contribute significantly to the development of a cost-effective yet highly efficient SPR biosensor platform for detecting EOC. The review is well organized and written on SPR for EOC, providing more useful information for detecting EOC and bringing benefits to EOC patients. It is, therefore, acceptable.

Answer:Thank you for the positive comments.

CHANGES WITHOUT MENTIONED IN COMMENTS

  1. Deleted/RemovedWe have removed the following lines: 43-45 and 45-46, as part of the improved writing regarding epidemiology and therapeutic approaches.

It is often referred to as a "silent killer" because it frequently goes unnoticed until it has already progressed aggressively from an early to an advanced stage, typically within one year.

line 43-45

The majority of EOC patients, more than 70%, receive a diagnosis when the cancer has already reached an advanced stage[3, 5]

line 45-46

  1. Amend at Simple Summary: line12-13

We have improvised the “simple summary” in line 12-13.

Simple Summary: This review significantly contributes to the research community by providing insights into the role of serum-based biomarkers for the diagno

Reviewer 3 Report

Comments and Suggestions for Authors

In this detailed review, the authors, provided a detailed overview of the recent advancements in immunoassay detection, employing multiplex technology and SPR biosensors to identify CA124 and HE4 biomarkers for diagnosing epithelial Ovarian Cancer. Please see comments and observations for improving the manuscript which I consider suitable for publication in Cancers.

MAJOR COMMENTS

1.     For completeness of information, I suggest including an additional introductive section on ovarian cancer epidemiology and therapeutic approaches.

2.     Besides the diagnostic biomarkers, it would be helpful to include a couple of words of the circulating prognostic biomarkers in epithelial Ovarian Cancer. The prognostic utility of mentioned biomarkers such as CA125 should be emphasized in the text, as the authors are prone to mention their diagnostic utility, only

MINOR OBSERVATIONS

1.     All abbreviations should be carefully checked. For instance, “limit of detection (LOD)” is repeated 2 times in lines 26 and 31. All names/words should be detailed at the first mention followed by their abbreviation, while in the subsequent sections abbreviations should always be used. Also, several acronyms are not detailed, e.g., line 145 “FDA”. Please revise the review ms accordingly.

2.     Similarly numerous typos should be revised. For instance, references should be separated from the last word of the sentences

3.     I suggest avoiding abbreviations in the sub-head titles, e.g. line 102, 135 etc. Periods should also be avoided, e.g., lines 343 328 and others

4.     Among the bioactive circulating compounds listed in line 105, I would include non-coding RNAs. This notion and both supporting references should be included. In fact, several miRNAs have been repeatedly identified as epithelial Ovarian Cancer diagnostic biomarkers PMID: 37238740. This is an important information that deserve attention

5.     These recently published works in the field of epithelial Ovarian Cancer diagnostic approaches should be included. PMID: 35223789.

6.     The quality of figure 2 should be improved. Other figures are fine and highly detailed and informative. Tables are well designed, as well

Comments on the Quality of English Language

 The Quality of English Language is good

Author Response

We would like to thank the Editor and Reviewers for their constructive comments and suggestions. Highlighted in yellow are our responses to the queries and corrections made in the manuscript.

REVIEWER #3 COMMENTS AND AUTHOR’S RESPONSES

Minor editing of English language required

Answer: The Manuscript has been proof-read by native English-speaking personnel.

In this detailed review, the authors, provided a detailed overview of the recent advancements in immunoassay detection, employing multiplex technology and SPR biosensors to identify CA124 and HE4 biomarkers for diagnosing epithelial Ovarian Cancer. Please see comments and observations for improving the manuscript which I consider suitable for publication in Cancers.

Answer: Thank you for the positive comments

MAJOR COMMENTS

  1. For completeness of information, I suggest including an additional introductive section on ovarian cancer epidemiology and therapeutic approaches.

Answer: We have included the therapeutic approaches in line 40-53

EOC accounts for 90% of all ovarian cancer and displays significant variation in terms of its appearance, clinical characteristics, molecular makeup, histologic subtype, and chemotherapy sensitivity, therefore, it has an impact on the prognosis of ovarian cancer [3-6]. EOC primarily manifests in older women, with the average age of diagnosis being 63 years, compared to younger women [7]. Despite a lack of screening methods, the rise in survival rates for ovarian cancer is largely credited to aggressive surgical debulking that includes hysterectomy and bilateral salpingo-oophorectomy, a standard surgical treatment for EOC and often combined with chemotherapy, specifically the conventional carboplatin-paclitaxel regimen [6, 8]. Recently embraced treatments, specifically Poly ADP Ribose Polymerase (PARP) inhibitors, have proven to be effective for women with EOC, especially those with BRCA mutations [5, 9]. Despite the initial effectiveness of these inhibitors, many patients eventually undergo an elevation in deoxyribonucleic acid (DNA) damage, resulting in resistance to PARP inhibitors and, consequently, an elevated risk of mortality [9].

Answer: We have updated the ovarian cancer epidemiology as follows in line 54-63

          American Cancer Society estimates for ovarian cancer in the United States for 2023 approximately 19,710 new cases in 2023 (1% of all new cancer cases) and the estimated number of deaths in 2023 is 13,270 (2.2% of all cancer-related deaths) [10, 11]. Despite these advances, the Surveillance, Epidemiology, and End Results (SEER) data indicate that the 5-year relative survival rate for ovarian cancer from 2013 to 2019 is at 50.8%  [10]. In contrast, a small percentage of patients are diagnosed with stage I EOC, and 78% of women diagnosed at this stage survive for at least one year [8, 12]. Hence, timely identification of early-stage epithelial ovarian cancer (EOC) is pivotal for enhancing prognosis. It is essential to address the non-specific clinical signs of EOC and discover biomarkers capable of signaling the existence of preclinical or early-stage tumors, while also offering prognostic insights [4, 13].

  1. Besides the diagnostic biomarkers, it would be helpful to include a couple of words of the circulating prognostic biomarkers in epithelial Ovarian Cancer. The prognostic utility of mentioned biomarkers such as CA125 should be emphasized in the text, as the authors are prone to mention their diagnostic utility, only

Answer: We have updated the prognostic and diagnostic utility as follows in:

line183-215

The diagnostic effectiveness of CA19-9, CA125, and CEA in preoperative serum was assessed through ELISA, ECLIA, and Magnetic Bead assay methodologies, highlighting their utility in predicting and distinguishing various types of tumors [46]. Moreover, CA125, HE4, and CA19-9 play a role in prognosticating patients with epithelial ovarian cancer (EOC). A meta-analysis, encompassing 23 studies with a total of 10,594 women diagnosed with EOC, revealed that a higher pre-treatment serum CA125 level, independent of FIGO stages and treatments, was correlated with poorer overall survival (OS: HR=1.62, 95% CI=1.270-2.060, p<0.001) and progression-free survival (PFS: HR=1.59, 95% CI=1.44~1.76, p<0.001). Consequently, serum CA125 stands out as a reliable indicator for predicting the risk of EOC disease progression  [47]. Additionally, preoperative HE4 emerges as a promising prognostic biomarker in EOC, especially in serous tumors, with elevated preoperative plasma HE4 levels (≥277 pmol/L) showing a significant association with increased EOC mortality (adjusted hazard ratio (aHR): 1.90; 95% CI: 1.09–3.29) [48].  

Furthermore, Rong and Li discovered that the timely normalization of HE4 and CA125 levels in the early stages could serve as an indicator for platinum response and prognosis in ovarian cancer patients. Thus, monitoring these biomarkers in combination throughout the initial chemotherapy phase could provide valuable insights into predicting platinum sensitivity and the likelihood of progression and relapse [49] Another study uncovered the role of CA19-9 as a prognostic biomarker. Zhu et al. found that combining postoperative CA19-9 and CA-125 appeared to have significant clinical value for prognosis in patients with ovarian clear cell carcinoma (OCCC) after initial debulking. Elevated postoperative CA19-9 was identified as an independent risk factor for both recurrence-free survival (RFS: HR, 5.0; P = 0.005) and overall survival (OS: HR, 1.1; P = 0.035) in patients with normal postoperative CA-125 levels[50].

Although CA125, HE4, and CA19-9 are standard detection markers for prognosis, there are a few new biomarkers used to predict the progression of disease in advanced EOC include Bikunin, Osteopontin (OPN), and Creatine Kinase B (CKB) [5]. These biomarkers are predominantly utilized in clinical diagnosis and prognosis, aiding in predicting the disease stage and monitoring chemotherapy response through ELISA. A recent review has outlined nearly two hundred prognostic biomarkers linked to EOC, establishing a robust groundwork for the exploration of innovative treatment approaches [33, 51]. The identification of prognostic factors in EOC patients is vital for devising optimal treatment strategies [33]. This involves establishing an algorithm to monitor various target cohorts, necessitating extensive databases.

line274-277

Multiplex immunoassays offer a more precise and comprehensive evaluation of the disease by measuring the levels of multiple biomarkers in serum samples and emphasizing the ability to predict the progression of EOC[53].

line 822-825

The emergence of SPR sensors offers the potential to enhance the use of post-operative biomarkers for clinical prognosis and cancer treatments, thereby reducing expenses and improving healthcare outcomes.

MINOR OBSERVATIONS

  1. All abbreviations should be carefully checked. For instance, “limit of detection (LOD)” is repeated 2 times in lines 26 and 31. All names/words should be detailed at the first mention followed by their abbreviation, while in the subsequent sections abbreviations should always be used. Also, several acronyms are not detailed, e.g., line 145 “FDA”. Please revise the review ms accordingly.

All abbreviations have been checked

  1. Similarly numerous typos should be revised. For instance, references should be separated from the last word of the sentences

Answer: This has been corrected in manuscript

  1. I suggest avoiding abbreviations in the sub-head titles, e.g. line 102, 135 etc. Periods should also be avoided, e.g., lines 343 328 and others

Answer: This has been corrected in manuscript

  1. Among the bioactive circulating compounds listed in line 105, I would include non-coding RNAs. This notion and both supporting references should be included. In fact, several miRNAs have been repeatedly identified as epithelial Ovarian Cancer diagnostic biomarkers PMID: 37238740.

Answer: We have included. PMID: 37238740 in manuscript as follows,line 127-136 Recent studies underscores the importance of placing greater emphasis on exosomal MicroRNA (miRNA), which is extensively implicated in the development of the epithelial ovarian cancer tumor microenvironment (EOC-TME). This emphasizes its crucial role in both the initiation and progression of tumors [30, 31]. Several researchers have examined the expression profiles of miRNAs in serum samples from individuals with EOC, with the goal of identifying biomarkers for the condition [30, 32]. Although a specific study effectively identified certain circulating miRNAs consistently linked to epithelial ovarian cancer (EOC), there is a lack of consensus regarding the methodologies employed [32], and the costs associated with miRNA profiling are substantial [24]. Currently, there is no definitive biomarker available to predict the response to therapy, whether based on mRNA, protein, or DNA. None of these markers has received approval for clinical use [5, 33].  

  1. These recently published works in the field of epithelial Ovarian Cancer diagnostic approaches should be included. PMID: 35223789.

Answer: We have included. PMID: 35223789 in manuscript as follows,line 87-89

The current approaches for investigating EOC biomarkers linked to exosomes in human serum include flow cytometry, protein microarray, diagnostic magnetic resonance, nanoplasmonic sensing technology, and microfluidics [24].

  1. The quality of figure 2 should be improved. Other figures are fine and highly detailed and informative. Tables are well designed, as well

Answer: This has been corrected in manuscript

                    Multiplex technology

Key points 

Luminex technology

PEA technology

Priniciple

Bead-based Immunoassay

Proximity ligation and Amplification

Number of analytes

100

92

Sensitivity

Variable,

Depends on assay design,

Dilution factor of analyte

High sensitivity

Application

Protein biomarker analysis,

cytokine profiling

Protein biomarker analysis,

New biomarker discovery,

Drug development

Commercial availbility

Widely

Available from specific vendors

Limitation

·        Expensive biomarker panel/assay

·        Complexity and technical expertise

·        Cross-reactivity and specificit

·        Potential interference from the sample matrix

·        Limited customization panel

·        Requires extensive validation

The authors reviewed Recent Advances in Surface Plasmon Resonance (SPR) Technology for Detecting Ovarian Cancer Biomarkers. This manuscript is interesting and informative for the readers.

Answer: Thank you for the positive comments

CHANGES WITHOUT MENTIONED IN COMMENTS

  1. Deleted/RemovedWe have removed the following lines: 43-45 and 45-46, as part of the improved writing regarding epidemiology and therapeutic approaches.

It is often referred to as a "silent killer" because it frequently goes unnoticed until it has already progressed aggressively from an early to an advanced stage, typically within one year.

line 43-45

The majority of EOC patients, more than 70%, receive a diagnosis when the cancer has already reached an advanced stage[3, 5]

line 45-46

  1. Amend at Simple Summary: line12-13

We have improvised the “simple summary” in line 12-13.

Simple Summary: This review significantly contributes to the research community by providing insights into the role of serum-based biomarkers for the diagno

Reviewer 4 Report

Comments and Suggestions for Authors

The authors reviewed Recent Advances in Surface Plasmon Resonance (SPR) Technology for Detecting Ovarian Cancer Biomarkers. This manuscript is interesting and informative for the readers. The reviewer has enjoyed reading this review and has made some minor comments to improve the manuscript. Please address the following comments before publication.

Line 15: CA124 should be CA125.

Line 41: Please cite the most recent statistical data.

Reference lists: please check all references carefully.

Comments on the Quality of English Language

English is fine.

Author Response

We would like to thank the Editor and Reviewers for their constructive comments and suggestions. Highlighted in yellow are our responses to the queries and corrections made in the manuscript.

Reviewer #4 comments and Author’s responses

Minor editing of English language required

Answer: The Manuscript has been proof-read by a native English-speaking personnel.

The reviewer has enjoyed reading this review and has made some minor comments to improve the manuscript. Please address the following comments before publication.

Answer: Thank you for the positive comments.

  1. Line 15: CA124 should be CA125.

Answer: This has been corrected in manuscript

  1. Line 41: Please cite the most recent statistical data.

Answer: We have included the statistical data in line 54-63

American Cancer Society estimates for ovarian cancer in the United States for 2023 approximately 19,710 new cases in 2023 (1% of all new cancer cases) and the estimated number of deaths in 2023 is 13,270 (2.2% of all cancer-related deaths) [10, 11]. Despite these advances, the Surveillance, Epidemiology, and End Results (SEER) data indicate that the 5-year relative survival rate for ovarian cancer from 2013 to 2019 is at 50.8%  [10]. In contrast, a small percentage of patients are diagnosed with stage I EOC, and 78% of women diagnosed at this stage survive for at least one year [8, 12]. Hence, timely identification of early-stage epithelial ovarian cancer (EOC) is pivotal for enhancing prognosis. It is essential to address the non-specific clinical signs of EOC and discover biomarkers capable of signaling the existence of preclinical or early-stage tumors, while also offering prognostic insights [4, 13].

Reference lists: please check all references carefully.

Answer: Thank you for highlighting it, it has been checked.

CHANGES WITHOUT MENTIONED IN COMMENTS

  1. Deleted/RemovedWe have removed the following lines: 43-45 and 45-46, as part of the improved writing regarding epidemiology and therapeutic approaches.

It is often referred to as a "silent killer" because it frequently goes unnoticed until it has already progressed aggressively from an early to an advanced stage, typically within one year.

line 43-45

The majority of EOC patients, more than 70%, receive a diagnosis when the cancer has already reached an advanced stage[3, 5]

line 45-46

  1. Amend at Simple Summary: line12-13

We have improvised the “simple summary” in line 12-13.

Simple Summary: This review significantly contributes to the research community by providing insights into the role of serum-based biomarkers for the diagnosis of ovarian cancer.